# Broad and Selectively Deep: An MRMPM Paradigm for Supporting Analysis

**Paul K. Davis**

RAND Corporation and Pardee RAND Graduate School, Santa Monica, CA 90407-2138, USA; pdavis@prgs.edu

**Abstract:** This paper discusses challenges for M&S if it is to be increasingly important to decision aiding and policy analysis. It suggests an approach that—from the outset of a policy analysis project—incorporates M&S of a varied resolution with the intent that (1) the results of analysis will be communicated with a relatively simple model and corresponding narrative that scans the system problem in breadth, having been informed by richer modeling, and (2) the broad view is supplemented by the selective detail (zooms) and selected change of the perspective as needed. This is not just a matter of "dumbing down" communication, but a matter of thinking about both forests and trees from the outset and about designing analytic tools accordingly. It will also enable exploratory analysis amidst uncertainty and disagreement, which is central to modern policy analysis and decision-aiding. All of this poses significant challenges for those who design and build M&S.

**Keywords:** modeling for policy analysis; multiresolution modeling (MRM); multiresolution; multiperspective modeling (MRMPM); multimodeling; families of models; base and lumped models; contextual abstraction; modeling and simulation; exploratory analysis; decision making under deep uncertainty (DMDU); robust decision making (RDM); qualitative modeling

## 1. Introduction

### 1.1. Motivation

Organizations often use relatively complicated models and simulations (M&S) for research and to support planning. A recurring problem is that the models may appear to leaders to be incomprehensible and insufficiently responsive. This greatly undercuts their value, as occurred in 2011 when the U.S. Department of Defense dissolved a large modeling group [1]. In this paper, I suggest methods to improve the substantive quality of work while greatly mitigating communication problems. An admonition for analysts is—from the outset—to use multi-resolution, multi-perspective modeling (MRMPM) with relatively simple models for seeing forests and communicating to leaders, and with relatively more complicated models for understanding deeper issues, while noting the consequences of alternative perspectives [2] and avoiding blunders [3].

Such an MRMPM approach should also include human gaming and human-in-the-loop simulation, which illuminate matters obfuscated in usual computer modeling. When possible, a project should have a campaign plan for analysis that includes not only MRMPM, but a variety of such other sources of information as empirical analysis and discussion with experienced operators [4]. Further, it should confront the ubiquitous uncertainty and disagreement common to the wicked problems arising in policy analysis for complex systems [5]. The paper's suggestions are consistent with the themes of the movement for decision making under deep uncertainty (DMDU) [6].

### 1.2. Terminology

In this paper (rather than the literature on computer graphics), multiresolution modeling (MRM) is "building a single model, a family of models, or both to describe the same phenomenon at different levels of resolution" [7]. An earlier term was "variable resolution

modeling" [8,9]. MRM ordinarily involves families of models. MRMPM is similar except that the model family also represents alternative perspectives [7]. Although not discussed further here, a model's resolution can be quite different with regard to time, space, objects, object attributes, and processes [9,10].

Considerable work has been accomplished with multiresolution modeling [11] by one term or another, as with multilevel modeling in Rouse's work [12,13]. For example, Moon has discussed the often-puzzling relationships among abstraction, resolution, and fidelity [14]. Bettini, Jajodia, and Wang have discussed issues of time granularity in considerable detail [15]. Some have explored MRM issues in connection with the DEVS formalism for simulation [16,17]. In a recent work, Rabelo and colleagues reviewed MRM issues integrating live, virtual, and constructive simulations [10]. Hadi used MRM in a detailed guide for traffic planning [18]. Zeigler and co-authors discussed MRM in their third-edition M&S textbook [19]. As their text describes, complex systems have multiple components, each of which can be described by a *base* model and a *lumped* model that is an abstraction of the first. They discuss the criteria for consistency adequate for a specified purpose and context. See Chapters 15 and 22, which are an online lecture [20] and a short article relevant to defense work [21].

The alternative perspective aspect of MRMPM does not seem to have been much addressed in the literature, but it is related to the discussion of multifaceted modeling [19] and some of the literature on multimodels. A multimodel is a modular model with sub-models that together describe the behavior of a complex multi-phased system [11]. Multimodels reflect the common need for a variety of abstractions in and across applications [22]. Modeling alternative perspectives is especially important in the social sciences where many crucial issues involve values. Being able to generate the appropriate range of alternative perspectives may require diversity in the analytic team and organizational license to acknowledge such alternatives [2].

Another term worth discussing at the outset is "validity." Although organizational leaders sometimes demand validated models, it makes no sense to talk about a model's validity in the abstract, a point emphasized long ago by Forrester and discussed in the textbook on system dynamics [23]. As recognized by the U.S. Department of Defense for decades, a model's validity should be assessed for a particular purpose in a particular context. Further, it is now recognized that "validity" should be seen as a multi-dimensional concept: a model's validity should be assessed separately for each of the following dimensions [24–26].

(1) Description;
(2) Cause–effect explanation;
(3) Postdiction;
(4) Exploration;
(5) Prediction.

The members of a multiresolution family will typically have very different realms of validity. For example, a machine-learning model may predict new data from the same stable system but will usually provide little or no cause–effect explanatory power. In contrast, simulation models (e.g., system dynamic models) are often valuable for description, explanation, and/or exploration, but with little or no predictive power (if only because their input data are so uncertain).

Other systematic discussions of validation take a different tack and focus more on means for assessing aspects of validity, notably testability, as in a paper that is careful in discussing the balance between simplicity and veridicality [27]. The U.S. Department of Defense, among other agencies, has an official article on the related issues of verification, validation, and accreditation [28]. Rouse discusses the many different functions of models in human-centric private- and public-sector applications. He touches upon assuring that a multilevel model is a good enough approximation for the application [29].

### 1.3. Outline of What Follows

With this background, this concept paper unfolds as follows. Section 2 identifies ten major themes derived from my experience over the years. They do not purport to be tidy or the result of a formal and comprehensive literature review. Section 3 discusses a few past cases that tend to confirm the themes. Section 4 presents the conclusions, including some of the many challenges for those who develop M&S concepts, methods, and tools.

## 2. Themes

### 2.1. Analytic Support for Policy Analysis Should Provide Breadth, Selective Depth, and Selective Ability to Change Perspective

By and large, strategic decisionmakers need a top-down synoptic view of how their actions may affect the system. That is, they require breadth in the same way that a system designer needs a view of the whole. That said, some assessments and decisions depend on relatively detailed matters that cannot be taken for granted. Selectively, then, analytic support requires depth adequate to illuminate the troublesome issues. Similarly, analytic support should include the ability to discuss a problem area from selected different perspectives. The word "selected" is crucial because no study or study group can be comprehensive. Table 1 lists some of the many alternative ways one might represent a system. The intent is merely to illustrate that models for policy analysis may need to address such differing views.

**Table 1.** Illustrative alternative perspectives for which modeling might be desired.

| Type Contrast | Left Side of Continuum | Right Side of Continuum |
|---|---|---|
| Control | Centralized | Distributed |
| Approach to knowledge | Empirical (neopositivist) | Cause–effect theory as well as empirical observation |
| Comprehensiveness | Component focus | System focus |
| Metaphysical approach | "Western" | "Eastern" |
| Understanding | Actors operating in an environment | Structures and processes shaping actors and their choices (constructivism) |
| Economics | Rational-actor economy | Economy with actors having bounded rationality at best and some irrational behaviors |
| Economic lens | Socialism | Free enterprsie |

### 2.2. Providing Breadth, Depth, and Variation Is Best Done with MRMPM Model Families

It follows that a project should plan for MRM and MRMPM from the start. This means having models that can accept inputs at different levels of detail, rather than "selective viewing" (generating lower resolution outputs upon demand using a single higher-resolution model) [8]. The need for this becomes manifest when we realize that higher-level reasoning—ours as well as that of policymakers—amounts to using a lower-resolution model. Doing so is essential for sense-making [7,8].

In some cases, a single model can zoom to a greater or lesser resolution (*hierarchical variable resolution modeling* or IHVR) [30]. More typically, we need different models for different resolutions and different perspectives. Ideally perhaps, we might develop family members jointly to understand how to relate them, i.e., how to aggregate, disaggregate, and map among them. Realistically, as with human families, the members of a family may develop over time without joint design and may not "play together" well without expert adult supervision. A military example is the U.S. Air Force using both a mission-level model (*Brawler*) for air-to-air combat, including pilot styles and tactics, and a campaign-level model (*STORM*) in which air combat is much more aggregated. Only expert users can

use the former to estimate the parameters sensibly for use in the latter, and then only for the context of a particular application [31]. An epidemiological example would be expert practitioners relating individual and population-level disease models [32].

Representing alternative perspectives in a family of models is more unusual, except perhaps for using rotational coordinates as an alternative perspective in physics. As an example, options for an international agreement might be assessed by a probable effect on per-capita GDP or, instead, for effects on the structure of society and the health of the middle class. Otherwise, one analysis might purport to compare options by a weighted sum of scores, while another might rule out any option with bad effects on any of several considerations (e.g., the continued viability of a nation's fishing industry). Analysts can sometimes represent perspectives with different objective functions, perhaps with the non-linear weighting of criteria [33–35]. Nonlinearities often arise in strategic decision making because success may depend on *all* of a system's critical components possessing threshold levels of effectiveness or, e.g., because the political environment includes single-issue criteria that must be met.

*2.3. Useful Lower Resolution Models May Not Be Straightfoward Aggregations from More Detailed Model Isomorphic Relationships Are Not Required*

In thinking about base and simple model pairs, as suggested by Zeigler [20], it is useful to distinguish between (1) lumping achieved by isomorphic aggregation from the base model's structure and (2) using a simplified model that mimics aggregate behavior adequately, but whose structure has no obvious relationship to the more detailed model. Aesthetically, the former has always seemed preferable, but the latter is common and important.

Some may see it as natural to start with a relatively detailed model and then generate a lumped version by aggregating. This, however, often proves difficult, complex, and hard to understand. In contrast, simplified models may arise independently, e.g., by the empirical discovery of scaling laws, by intuitive leaps, or by heroic assumptions that someone familiar with details might be loath to make. Some examples familiar in both physical and social modeling involve linearity. There may be myriad reasons for aggregate behavior to be nonlinear, but linear approximations are often surprisingly good [36].

Another way to proceed is to start with a relatively simple model and add layers of detail to it as seems useful. This is apt to clarify morphisms between simple and more detailed models, but it may tend to oversimplify (and misrepresent) the more detailed phenomena. If one began with a simple rational, analytic economic model and then added detail, one might not postulate incorrectly a well-behaved and rational microlevel, rather than representing the more chaotic microscopic behavior of real systems (including actual human reasoning).

Interestingly, we humans are superb at shifting seamlessly between simplifications as circumstances change (we hardly notice doing so), whereas models are often not so good at doing so. In military work, force ratio models proved remarkably useful for decades, but only by practitioners who would quickly adjust the force ratios depending on circumstances [37,38]. In epidemiology, SIR and SEIR models have long proven valuable, albeit with the necessary adaptations [39].

The theme here is that in generating a family of models, we should not hesitate to contemplate "simple" models that at first blush appear simplistic. Perhaps the data will demonstrate that they are better than expected and perhaps in-depth thinking will reveal why.

As an example of why this matters technically, consider that the approximate validity of lumpability is often said to depend on the approximate uniformity and indifference, as when all the agents of a system's components behave the same and do not affect each other in ways that distinguish among them. Consider, however, a model F(X, t) with a state vector X that may have 100 or 1000s of components, such as soldiers in combat. The soldiers have varied abilities and may variously fight singly or in groups with small or large groups of enemy soldiers. In other words, microscopic combat may not be uniform or

indifferent. However, the overall outcome may be driven by only one or a few aggregate variables, such as the initial force ratio and perhaps a relative will to fight. Or, conversely, the aggregation of microscopic dynamics may *not* be so simple. The microscopic simulation may, near a tipping point of relative prowess, show a bifurcation into cases in which one or the other side wins decisively, rather than an outcome in which the sides mutually exhaust themselves into stalemate. This kind of trajectory splitting is common in the study of complex adaptive systems (CAS).

Using a COVID-19 example, individual-level sequences of infection, suffering, and recovering may vary across individuals and random events, but none of this may matter if sufficient mixing occurs. In contrast, if groups have different vulnerabilities and infection mechanisms, and do not mix very much, the disease may generate distinct populations. Averaging across populations may then be misleading. Naïve analysis might expect the disease to die out in a month or two, whereas reality might see disease disappear in one group, persist longer in another, and then manifest itself in cycles of disease. The prediction of cycling was one of the "wins" in the modeling of the COVID-19 pandemic. Such cycling is also not unusual in the modeling of complex or complex adaptive systems [23,40].

### 2.4. Within a Family, Models Should Be Cross-Calibrated for Mutual Consistency

A long-standing myth is that lower resolution models should be calibrated based on the runs of higher-resolution models. The myth may have been the result of analogies to physics where, e.g., classical thermodynamics can be derived from statistical mechanics, which can in turn be derived from quantum statistical mechanics. When drawing that analogy, however, it is often not appreciated that the parameter values of aggregate phenomena are usually obtained empirically.

The upward calibration approach might make sense if the detailed models and data (see top of Figure 1) were accurate and certain. Unfortunately, in application areas (as distinct from the idealized universes of physics theory), models at different levels of detail are flawed and their input data are uncertain [41]. A better approach for the M&S community is to seek mutual consistency across all levels using all the available empirical data [42]. That is, the bottom depiction in Figure 1 is more appropriate for the military example used: information flows upward, downward, and sideways, connecting information at all levels.

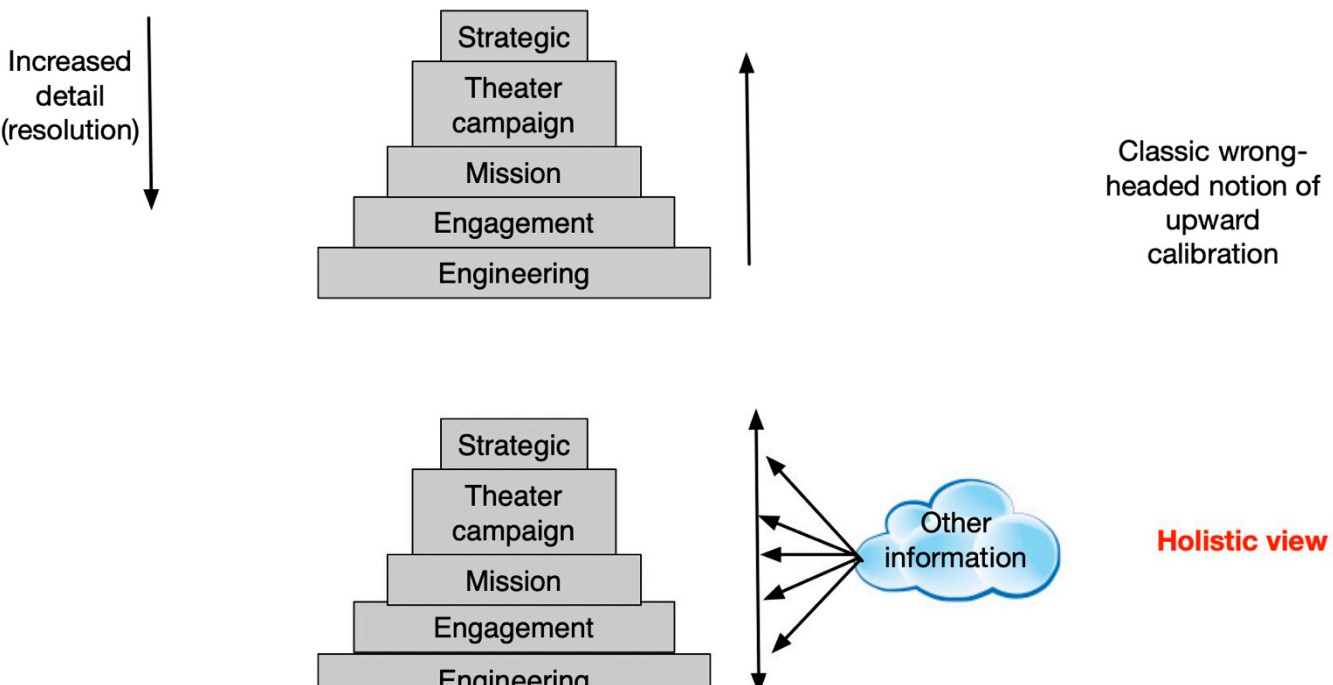

**Figure 1.** Objective: achieving consistency across combat models of varied resolution.

To illustrate significance, consider that many detailed models, coupled with their databases, predict aggregate system behaviors that are far more efficient than real-world aggregate empirical data. Often, the detailed models lack processes corresponding to frictions or inefficiencies. For example, when Russia invaded Ukraine in 2022, many models had predicted a quick victory. They did not allow for such factors as the relative will-to-fight of combatants, the relative prowess of the sides' generals, and systemic corruption affecting many aspects of the Russian army's quality.

As a second example, some early models of the COVID-19 pandemic assumed rational behavior on the part of populations, such as a quick and full acceptance of vaccines. Even if they had a parameter for vaccine hesitation (as did the influential Imperial College London model of the COVID-19 pandemic [43]), the model did not anticipate vaccine resistance correlating to political schisms and tribalism.

The idealized notion of upward calibration from a definitive detailed model to simpler aggregate models has always been misguided, even as an idealization. It may be a zombie idea, meaning it is one that will rise again and again.

### 2.5. Motivated Meta-Modeling Can Help with Cross-Calibration and Data Analysis

Operations researchers have long used response surfaces or meta models, often simple analytical expressions that predict approximately the same outcomes as a more complicated model regarded as authoritative. Such models, however, often provide no conceptual music. They are just regressions with various mysterious terms and coefficients.

A more attractive approach is to think about what simplified behavior might look like based on plausible but perhaps heroic assumptions, such as integrals being approximated by a representative value of the integrand times the width of the integration domain, a single chemical or social process dominating the problem, economically rational behavior, exponential decay of some property, or steady-state circumstances. If the resulting analytic expression makes sense dimensionally and conceptually, then the result can be tested against empirical data or against systematic runs of a more authoritative model. Perhaps the simplified model explains results well, albeit with an empirical multiplier and an empirical error term; if so, then the simplified model also provides the rough causal explanation that is so crucial to narrative and communication.

A premier example which has emerged over the decades is the radar equation (discussed in many places, including Wikipedia). Such a formula model can be used to specify elements of a regression, which is far better than merely having some coefficient values for a regression that seems to fit extant data for unknown reasons. James Bigelow and I called this approach *motivated metamodeling* [44]. When introducing the approach, we assumed that it was common. Instead, it seems uncommon, especially by data-driven analysts who talk about "letting the data speak" or those wedded to detailed models. Motivated metamodeling could be a routine element of policy analysis projects connecting models and data. I note that testing a motivated metamodel in no way corrupts analysis as data-driven researchers sometimes claim; if the postulated form is not roughly right, the corresponding coefficient will turn out to be small (the postulate will have been falsified). In contrast, even though users may think that they are allowing "data to speak," typical regression analysis imposes the assumption of linearity (or of a polynomial structure).

### 2.6. The Capacity for Exploratory Analysis under Uncertainty Needs to Be Built in from the Outset

Almost everyone agrees on the need for sensitivity analysis or even its cousin on steroids, exploratory analysis, that explores outcomes as all the important and uncertain parameters of a problem are varied simultaneously. In practice, however, studies usually do far less ambitious uncertainty analysis than intended, in significant part because it is difficult and tedious unless the groundwork has been laid from the outset in the modeling, programming, and analytical tools.

The methods for exploratory analysis and support for robust decision making (RDM) are hallmarks of considerable research in recent decades, associated with decision making

under deep uncertainty (DMDU) [4,6,45–50]. Such analysis is still the exception rather than the rule, but that should change. So the questions asked of models should also change; rather than asking "What if?" and running a simulation, we should routinely be asking "Under what circumstances and assumptions will this strategy succeed or fail" [51]? This is especially true in competitive domains, such as defense planning or commercial battles for market share, but it is true also in planning to avoid environmental and other disasters.

### 2.7. Modular Rather Than Monolithic Models Should Be the Rule

Organizations often focus on a single premier model as authoritative, usually a complex and monolithic model. Often, however, good analysis benefits from leaner, agile, and tailored work. Monolithic models do not lend themselves readily to this.

Individual groups develop model families that they learn to work with, but this usually requires a cadre with deep expertise and craft knowledge. Examples include the family of high-resolution simulations used by Matsumura, Steeb, and Herbert, and the work in several military service analytic teams such as the Air Force A-9. Some research groups have composed models for analysis that use agent-based modeling, system dynamics, and social science methods [52]. Some are taking a more modular approach to enterprise-level analysis, as described by Gallagher and colleagues [53].

### 2.8. Qualitative Models Can Be Structure and Subtantive

Although most members of the M&S community focus on quantitative applications, policy analysis typically involves the social sciences in which qualitative research is often particularly insightful. A project considering a family of models should not hesitate to address qualitative issues, especially since qualitative modeling can add significantly to the coherence of the work. One of the contributions qualitative modeling can make is to combine what otherwise might be carried along as competing social science theories that merely pass in the night without unification. More unifying qualitative theories may be highly structured, based on solid social science, and made quantitative for limited mathematical purposes (e.g., by the use of subjective Likert scales).

### 2.9. Interface Models

Sometimes it is necessary to build an "interface model" that maps the questions of policy analysis (sometimes questions from policymakers themselves) into the variables of an existing model. The policy questions often reflect perspectives different from those of model builders. Some examples are;

a.   What if we could double the rate at which assets are deployed?
b.   What if the adversary used strategy Y instead of strategy X?
c.   How would improved morale affect productivity?
d.   What if we allowed only vaccinated individuals to be in government workplaces?

Constructing such "interface models" can be nontrivial because if may require 1:n mapping (disaggregation) and the good ways for doing that may or may not be obvious. For example, policymakers may provide more resources, but the value of doing so may depend on how they are allocated (perhaps evenly, "optimally" with a mission in mind, by organizational clout, or by doctrine).

### 2.10. The Simplest Family Member May Be a Graphic or "Common-Sense" Argument

Readers of this journal are comfortable with mathematics. For both project-level discussion and up-the-chain communication, however, the ideas may best be expressed with a simple graphic, a short essay, or a $2 \times 2$ logic table such as those of which political scientists are notoriously fond. As one example, we may take a system view and note that "we need to assure that all the system's critical components are satisfactory." Heads will nod in agreement. However, the significance of this may only be internalized when people see charts or hear stories where disaster occurs because of a single failed component (a

horseshoe nail or an angle-of-attack sensor on a Boeing 737-Max). Thus, it may be helpful to demonstrate such points with simple computational models, even, perhaps, as simple as:

If efectiveness depends on critical components $\{A_i\}$, then if any $A_i$ is below its critical value $A_{io}$, effectiveness is 0. Otherwise, effectiveness is a weighted sum. In pseudocode, this might be

If Min $(A_i-A_{io},i)< A_{io}$ Then 0 Else Sum$(W_i*A_i,i)$

As another example of how simple summary graphics can have a big impact, we might recall the early COVID-19 modeling by Imperial College London [43]. It generated graphics predicting the collapse of the healthcare system if the pandemic's infection rate was not greatly reduced (e.g., Figure 2). These projections led to measures such as directives to maintain space between people, avoid large assemblies, and enact temporary lockdowns. Although the original analysis was criticized and was subsequently found to have made some incorrect estimates [54], it was extremely influential.

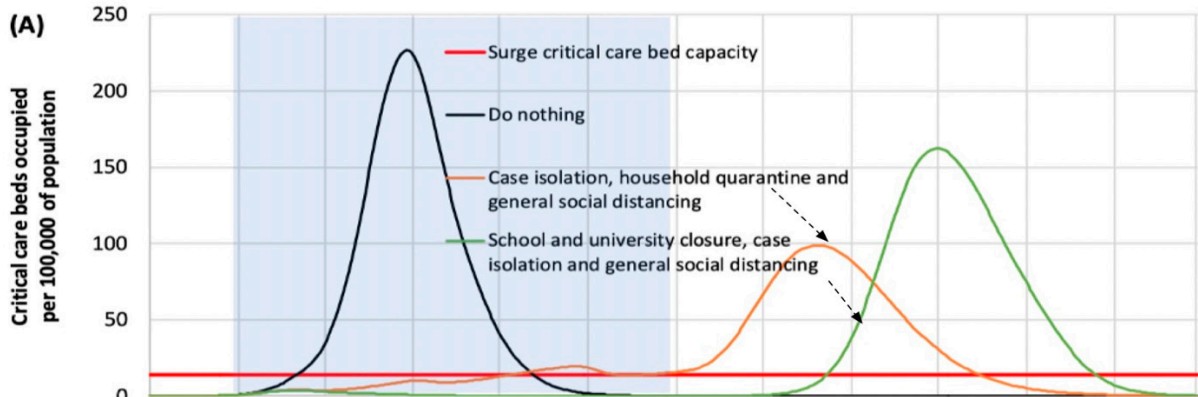

**Figure 2.** Early projection of COVID-19 patients in critical care with and without mitigation. Source: [43].

## 3. Cases

The cases that follow are brief summaries of some experiences over the years that influenced the themes discussed in Section 2.

### 3.1. Radiation from High-Altitude Rocket Exhaust Plumes

During the 1970s, the U.S. Defense Advanced Research Projects Agency (DARPA) studied radiation from high-altitude rocket exhaust plumes. Its program included aerodynamic simulations with embedded models of chemical reactions, shock-tube testing, testing from different platforms, and other components. As the program approached conclusion, however, the results from the component studies were inconsistent with each other and the empirical data, which was depressing. Fortunately, I was able to construct a one-liner "formula model" derived from the physics (along with heroic assumptions) to explain qualitatively what sensors might observe as a rocket flew through a high altitude with some calibration based on simulations. Although originally intended as a rough heuristic, the model provided coherence to the research project and explained the previous contradictions. As a bonus, it proved to be accurate as a scaling law. In retrospect, having had the formula model from the outset would have affected many elements of the research program. Much of the research of this program (although not the model) was described in an open publication by Fred Simmons [55].

### 3.2. The Military "Halt Problem" of the Eary 2000s

In the 1990s, the U.S. military was beginning to assimilate precision-strike capabilities to stop (i.e., "halt") invasions of friendly countries. The effectiveness of precision fires depended on scenario-driven variables such as the availability of shooters, basing, the

attacker's strategy, terrain, and air defenses. Thus, when complex joint campaign models were used to assess halt campaigns, the results depended on scenario details, which led to fierce debate among factions for and against the new technology (in part, battles about budget share).

Once again, simpler models sharpened issues, allowed exploratory analysis across scenario space, and explained the results [56,57]. A simple analytical model organized a good deal of subsequent simulation (see appendix of [56]). The simple model was summarized with a single graphic showing how the key variables affected the halt time or halt distance (Figure 3). In retrospect, the Greek letters and matrix notation were unfortunate. Later models were expressed in admirably clear prose [57]. Interestingly, while some conclusions could be tested with more detailed joint campaign models, the more consequential issues could not; they required even more detail (see next case).

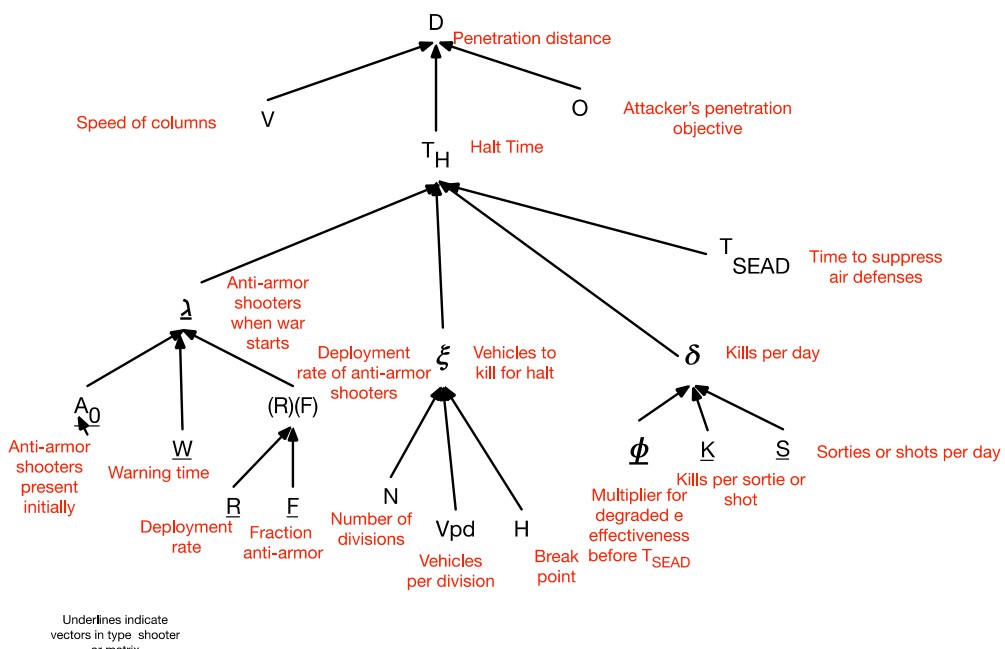

**Figure 3.** A simple graphic of the analytical halt problem model.

### 3.3. Effectiveness of Long-Range Precision Fires

Even when the long-range precision fires were available, as discussed above, their effectiveness was a mystery in the mid-to-late 1990s. Two entity-level simulation studies had differed by an order of magnitude in estimating the effectiveness of different scenarios [58,59]. Understanding this was very difficult because the entity-level *Janus* simulation and the accompanying scenarios were very detailed. *Janus* tracked individual mechanized vehicles as they moved through digitized terrain and engaged through line-of-sight targeting; or, in the man-in-loop simulation of the Army's long-range fire system (ATACMs), the process would include spotting a group of vehicles, transmitting information to an ATACMs battery, and launching ATACMs so that it might hit the advancing group at a projected future time and place.

Working with the previous authors, James Bigelow and I sought to untangle the mystery in a study treating the output from an entity-level simulation as empirical data. With fresh thinking about the physics, it became clear that even a "simple" model would need high resolution in a few respects, as suggested by Figure 4 [60]. This microscopic view of one aspect of the simulation was reflected in an otherwise simple simulation with such aggregate features as average spacing between armored vehicles and average vehicle speeds. For ATACMs to succeed, its impact would have to occur before the vehicles of the target packet entered a wooded or otherwise cluttered terrain, even if the ostensible

footprint of the ATACMs was huge. However, what did being "in the clearing" mean? Figure 5 shows the entity-level view. While an aggregate-level map might encode an area as a largely "open" area, much of such an "open" area is actually cluttered with objects (trees, houses and other buildings, and roads). This meant that to use the aggregate-level model, it would be necessary to use data on the size of open areas based on detailed work rather than aggregate maps. Thus, the exercise demonstrated the need for work at different levels of detail even though it also explained the previous mystery and provided a way to estimate the effectiveness of long-range precision fires for a wide range of conditions. Covering such a range with an entity-level simulation was out of the question.

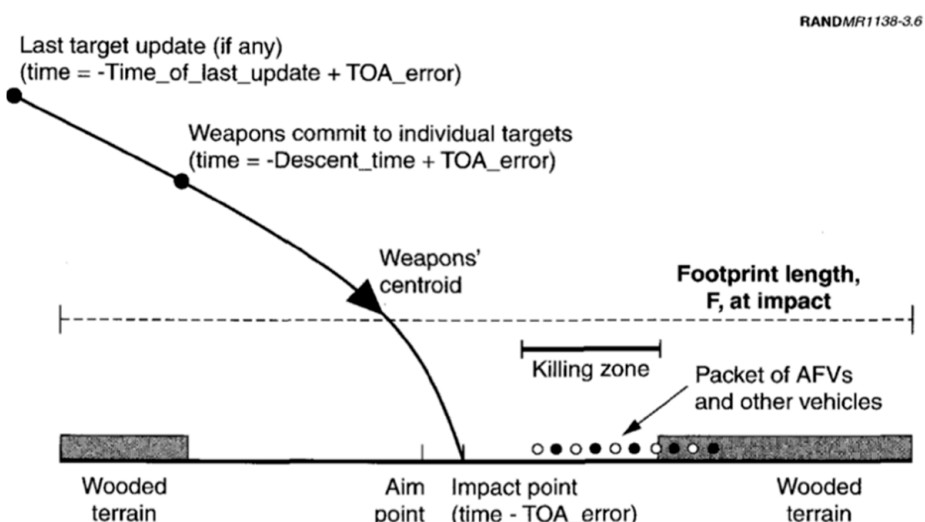

**Figure 4.** A simple but high-resolution physics model.

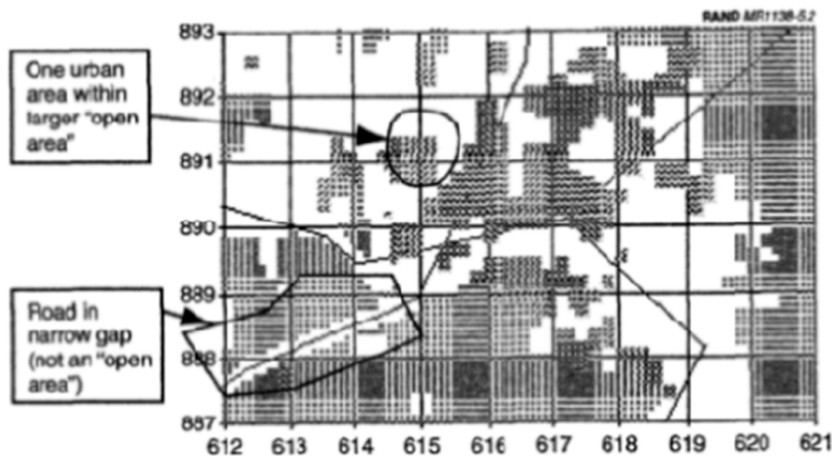

**Figure 5.** A high-resolution depiction of terrain. Source: Figure 5.2 of [60], p. 40.

### 3.4. Air Force Close Support of Ground Forces

Early in the 2000s, the U.S. Air Force leadership became concerned because field commanders in ongoing wars were not having their requests prioritized within the budgeted process. The Air Force system for planning and budgeting was dominated by issues related to high-cost, high-visibility systems that would typically not reach the field for years. Air Force leadership and the Secretary of Defense were frustrated by the budgeting system's tendency to focus on future wars to the exclusion of present wars. Richard Hillestad and I sought to illustrate in prototype what might be done to better diagnose current AF

problems, identify solutions, and explain the associated program needs. The prototype focused on close air support (CAS) to ground forces.

It would have been possible to approach the problem with the existing theater- and mission-level combat models, but doing so would have required a great deal of time gathering and negotiating detailed data on everything from weapon systems, forces, concepts of operations, and scenarios. Instead, from the outset, the team focused on the close-support mission and relatively simple modeling. Figure 6 describes the spreadsheet-level model used. It was developed by Paul Dreyer and programmed in *Visual Basic* on a spreadsheet. It included stochastic features and a number of decision processes. This was a useful level of detail, and the model was exercised for many cases.

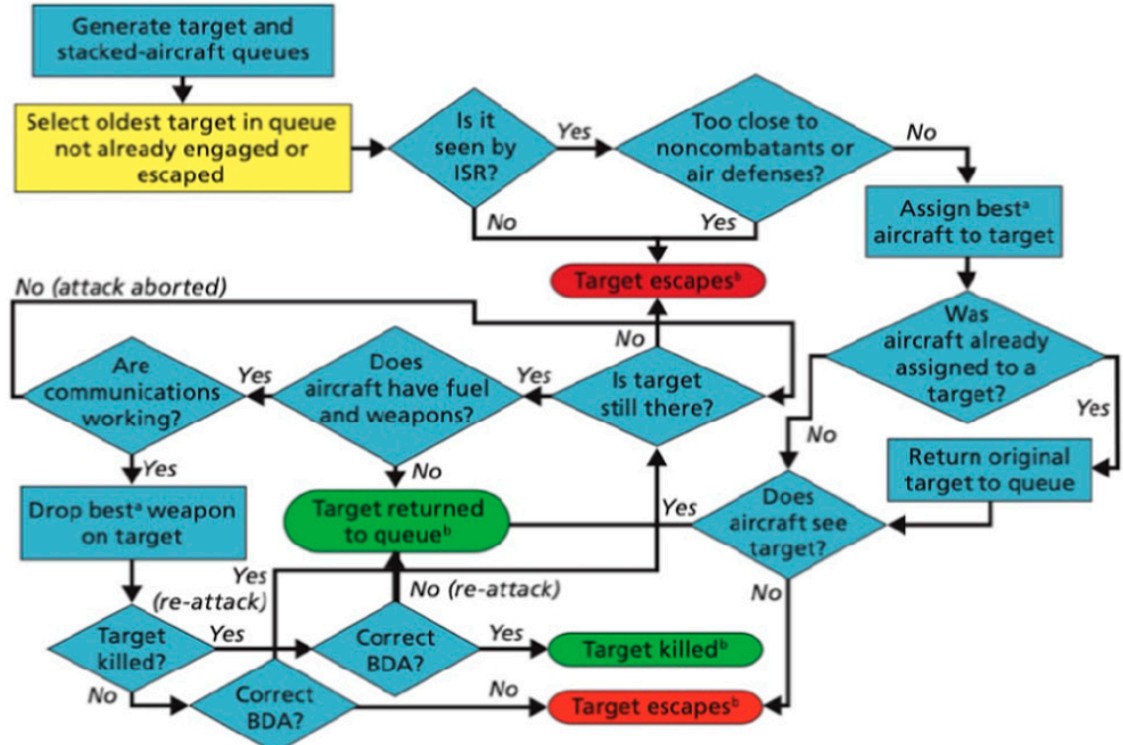

**Figure 6.** Schematic of the model. Source: [61].

After better understanding the issues, we developed a narrative understandable to general officers and policymakers. We constructed a different simple model, as indicated in Figure 7. Significantly, this model does not map neatly into that of Figure 6. Instead, it represents a different perspective of the problem. In this, the image is that CAS effectiveness would be low unless *each* of four separate factors were high: the probability of detecting and redetecting the target, the probability of the pilot receiving timely permission to engage, the probability that engagement would occur quickly enough to be militarily effective, and, finally, the probability of destroying the engaged target. From our discussion with pilots and other warriors, we had come to recognize that some of the major factors were in the command and control system. In practice, aircraft might not be allocated to the mission soon enough, enroute targeting decisions might be delayed (e.g., permission to engage), aircraft would hit their targets too late to help to the ground commander, and so on. Some of these system glitches were due to doctrine, which had evolved for the classic Air Force problem of attacking fixed targets. That doctrine tended to value measures such as targets killed rather than the timeliness of such kills.

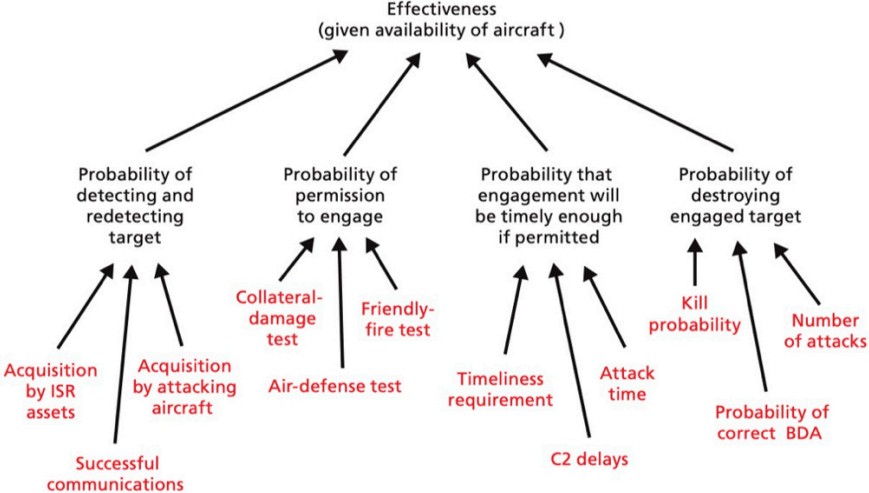

**Figure 7.** Schematic of a simpler depiction of the problem.

The spreadsheet model of Figure 6 described the phenomena well, but Figure 7 explained matters to the three-star sponsoring official, who appreciated the system view even though he was not himself a system engineer.

*3.5. Insights from the Social Science of Terrorism*

In 2007, the U.S. Department of Defense largely stopped the further M&S of terrorism and insurgency because the results to date had been disappointing. It stepped back to understand the underlying science, calling for a review led by social scientist Kim Cragin and me, which drew heavily on the qualitative social science literature and provided insight about cause–effect relationships rather than mere correlations. The result was a large, edited book [62], the themes from which were summarized in *factor tree* models. Figure 8 is an example of a factor tree from a follow-on study [63]. It is a multi-resolution qualitative model indicating the variables (factors) affecting public support for insurgency and terrorism.

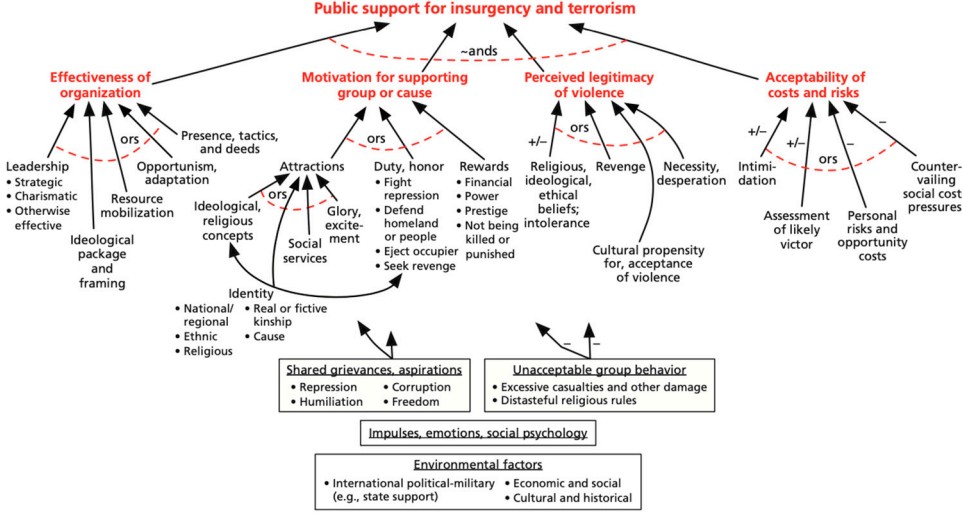

**Figure 8.** A factor tree model of public support for insurgency and counterterrorism.

Such factor trees appeared prominently in well-received briefings to general officers and larger audiences. Later, I showed, along with Angela O'Mahony, how a factor tree

model could be turned into a computational model, not a simulation, but rather a model "putting the pieces together" to predict the combinations of factors that would tend to generate public support for insurgency and terrorism or, conversely, the repudiation of the terrorism. Figure 9 is one summary graphics from [64,65]. In this depiction, the outcome is represented by the color or number of a table cell, rather than the vertical axis. Red (or the number nine) as a cell value shows high public support; green (or the number one) shows very low public support. The single graphic shows the results as a function of five factors. Regrettably, the graphic had to be manually generated using a combination of *Analytica, Excel,* and tedious manipulations. A tool for generating such displays would have been valuable.

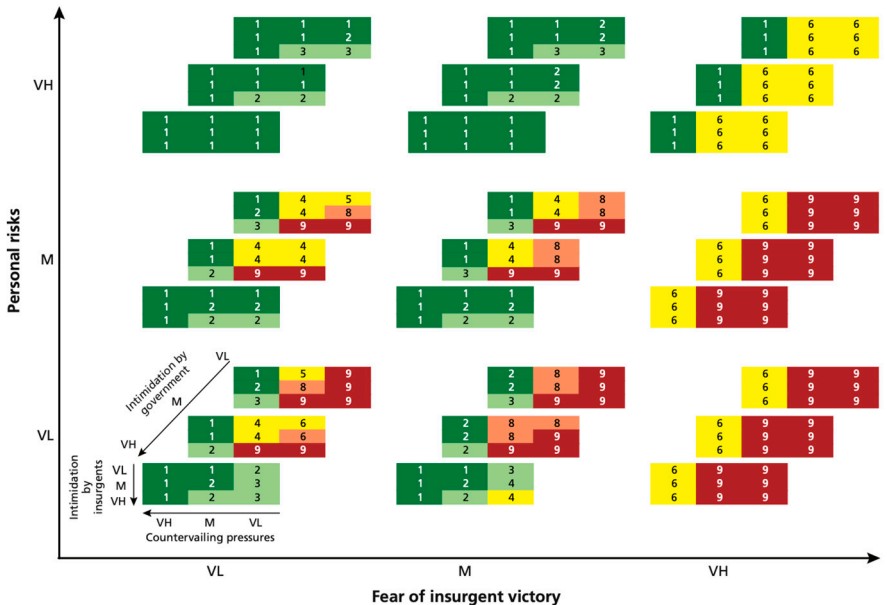

NOTES: The numbers shown are the values used internally in PSOT. Because low acceptability of costs and risks is good for the counterinsurgent side, the colors for 1, 3, 5, 7, and 9 are green, light green, yellow, orange, and red, respectively.

RAND *TR1220-S.2*

**Figure 9.** Summary results for a computational model built from Figure 8.

Figure 8 is a factor tree for a snapshot in time. However, what about dynamics? Ideally, the project would have had a family of models that included system dynamic simulations, agent-based simulations, empirical time series, and so on. We had none of that, but we *discussed* dynamical considerations based on the social science research in the text. This was important because the tree-like structure at a snapshot in time does not convey a sense of how the factors change over time and how they affect each other and become intertwined as they do so. At least one effort to undertake related modeling was undertaken by the U.S. Joint Staff in an attempt to understand in modeling language the implications of the then new Army–Marines counterinsurgency manual. The result was an interesting and insightful system dynamics model said to have been useful to people in the field attempting to understand the Iraqi and Afghan wars [66]. It was not, however, useful for prediction. Further, it was deemed to be too complicated to be helpful in most discussions.

Taken as a whole, the cases provide examples of all the themes of Section 2. Most obvious is the value of multiresolution modeling, but so also the cases demonstrated the significance of different perspectives, the use of motivated metamodels, designing *from the outset* for multi-dimensional exploratory analysis, the use of simple models not necessarily isomorphic with the more detailed models we used, the value of one-pager graphics or summaries, etc. Another lesson from these and other cases was that visual modeling was extremely effective for the design phase and for communicating to audiences of different technical sophistication. In much of this work, we used the *Analytica* modeling system because of its visual modeling and array-friendly declarative programming, which made

designing for exploratory analysis straightforward. When using models built in other languages, we often worked hard to construct and substantive accurate diagrams to explain their functioning.

## 4. Discussion

It is not the function of this paper to provide recipes on how to pursue the themes discussed in Section 2. Rather, my intent has been to suggest challenges and frontiers. My own view is that, while working across levels of resolution is currently an art, one in which good analysts are often skilled in without even being aware how special their talents are, much of the activity can be taught as a mixture of science and art. So, those developing the tools for M&S should also put more attention on generating tools to help in the challenges identified that will allow M&S to be more useful in policy analysis and decision-aiding.

As mere examples, I suggest the need for:

- Templates and tools to help specify and execute special-purpose aggregations in particular contexts. One such tool would generate experiments to inform local aggregations. Together, the tools might generate good enough heuristic rules and establish warning flags for when a heuristic is used out of range.
- Textbook advice on how to use "motivated metamodeling" routinely when analyzing data and how to use historical or other empirical data to test the models embodied in M&S when there is no straightforward mapping between what was measured and what is needed by the model.
- Textbook advice on conceiving and generating appropriately different model perspectives so model-based decision-aiding is not just conveying a single story but alternative stories reflecting different beliefs, values, and perspectives. The textbook advice might include examples of where such alternative perspectives have dramatic consequences, such as when urban planning looks different when viewed strictly in economic terms or in terms that value culture and urban character.
- Textbook advice on what Erica Thompson has called "Escaping from Model Land" to better engage the real world [2]. One aspect of doing so is the multiple perspectives previously mentioned.
- Computer languages or overlays to make multidimensional exploratory analysis routine and easy, to include using the kinds of tools associated with robust decision making and decision making under deep uncertainty (DMDU) (e.g., scenario discovery tools) [6,49,67].
- More emphasis in the M&S community on visual programming, whether in system dynamic languages such as *Stella* and *Vensim*, in using visual modeling platforms such as *Analytica* and *MATHLAB*, or in providing visual interfaces to models coded in languages such as *Python, R*, and *Java*. If experience should have taught us anything, it is that visual depictions are powerful in design, documentation, and communication.

**Funding:** This research received no external funding. It draws upon published material originally funded by the U.S. Department of Defense.

**Data Availability Statement:** Not applicable.

**Conflicts of Interest:** The authors declare no conflict of interest.

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
