# Peer review of "Broad and Selectively Deep: An MRMPM Paradigm for Supporting Analysis"

_information, doi:10.3390/info14020134_

Round 1

Reviewer 1 Report

General Comments:

This is an excellent paper. I am happy that the author took the time, after a long career, to reflect on modeling and simulation. I think many readers will benefit from this article. I have only a few minor edits and suggestions, as detailed below.

Detailed comments: 

Line 117: I like the table, but I offer the following suggestions:

  • Is "dichotomy" the same thing as opposite ends of a continuum? If so, I recommend changing to that nomenclature.
  • I would put "free enterprise" under End 2 and socialism under End 1. "Control" is much more distributed under free enterprise than it is under socialism.

Lines 123-124: I think this is a great point.

Lines 132-138: I'm glad you put in these examples of what "model families" would look like. I was a bit lost as to what they were until this point in the manuscript.

Lines 139-146: This explanation of multi-perspective models was less enlightening. Perhaps the author could come up with a couple of concrete examples?

Lines 154-155: I agree that simplified models are often more effective, but I don't see why the structure of such a model can't be representative of the system being modeled. Perhaps the author could expand on this a bit.

Lines 170-173: My own modeling experience is admittedly limited to system dynamics, but I have found that my practice moves from simple (yet effective) to larger and more detailed (but still showing the basic structure of the simpler, less detailed, model). I'm not sure how that relates to the author's point, but I agree that simple models can be extremely useful.

Lines 174-189: I like the example of soldiers in battle. I am familiar with CAS's, but not with the modeling of them. But I get the idea and this example helped.

Line 181: The word “by” appears twice in a row.

Lines 190-199: Again, I like the COVID example. I think what the author is getting at is that an SEIR model might produce one result while an agent-based model might produce a different one. Both would be useful in their own ways.

Lines 200-214: I really like this entire discussion. It clearly is a myth that lower-resolution models should be calibrated by higher-resolution models. The author's call for using data for calibration is obviously a better approach.

Line 224: Should be “vaccine-hesitation,” not “vaccine-hesitatio”

Line 250: Start a new paragraph at "A premier..."? It just seemed that you started a substantial new thought/discussion at this point, and formatting it in a new paragraph would help the reader.

Lines 298-300: Great point about the value of qualitative models.

Lines 305-320: I'm not entirely sure I understand the author's idea of "interface models," but if it means that models should be "operational," such that possible events/factors can be entered into them, then I wholeheartedly agree.

Lines 329-332: Great example about the value of "small leverage point" stories.

Reviewer 2 Report

Very interesting paper.

Important for future development around multi resolution Modeling and Simulation.

Maybe the following papers could be cited :

Bettini C, Jajodia SG, Wang SX. Time granularities in databases, data mining and temporal reasoning. 1st ed. Secaucus, NJ: Springer-Verlag New York, Inc., 2000.

J.F. Santucci, L. Capocchi, B.P. Zeigler, System Entity Structure Extension to Integrate Abstraction Hierarchy and Time Granularity into DEVS Modeling and Simulation, Simulation, Society for Modeling and Simulation International (SCS), July 2016. doi: 10.1177/0037549716657168

Moon IC, Hong JH. Theoretic interplay between abstraction, resolution, and fidelity in model information. In: Winter simulation conference, Washinton DC, USA, 8–11 December 2013, pp.1283–1291. New York: IEEE.

Baohong L, Kedi H. A formal description specification for multi-resolution modeling (MRM) based on DEVS formalism. In: 13th international conference on AI, simulation, and planning in high autonomy systems, Jeju Island, Korea, 4–6 October 2004, pp.285–294. Berlin, Heidelberg: Springer-Verlag.
